# Two Annulated Azaheterocyclic Cores Readily Available from a Single Tetrahydroisoquinolonic Castagnoli–Cushman Precursor

**DOI:** 10.3390/molecules25092049

**Published:** 2020-04-28

**Authors:** Elizaveta Karchuganova, Olga Bakulina, Dmitry Dar’in, Mikhail Krasavin

**Affiliations:** Institute of Chemistry, Saint Petersburg State University, 199034 Saint Petersburg, Russia

**Keywords:** Castagnoli–Cushman reaction, hydroxamic acid, heteroannulation, nitroarene reduction, tetrahydroisoquinolonic acid, indoloisoquinoline, dibenzonaphthyridine, cryptolepine

## Abstract

A novel approach to indolo[3,2-*c*]isoquinoline and dibenzo[*c*,*h*][1,6]naphthyridine tetracyclic systems was discovered based on switchable reduction of 2-methoxy-3-(2-nitrophenyl)-1-oxo-1,2,3,4-tetrahydroisoquinoline-4-carboxylic acid prepared via Castagnoli–Cushman reaction. Reduction with ammonium formate resulted in the expected selective transformation of the nitro group (thus providing access to substituted dibenzo[*c*,*h*][1,6]naphthyridine via cyclization and dehydrogenation). However, attempted reduction with sodium sulfide initiated a previously unknown reaction cascade including double reduction, cyclization, and decarboxylation, leading to formation of indolo[3,2-*c*]isoquinoline polyheterocycle in one synthetic step.

## 1. Introduction

During the last decades, the formal [4 + 2] cycloaddition of homophthalic anhydride (HPA) and imines (recently dubbed the Castagnoli–Cushman reaction, CCR [1,2,3]) has proven itself as a powerful tool for accessing tetrahydroisoquinolonic (THIQ) acids **1** (Figure 1), the promising scaffolds for the design of biologically active compounds. Recently, we reported [4] a practically simple approach to cyclic hydroxamic acids (*N*-hydroxy THIQ acids **1**, R^1^ = OH) via the CCR, which employed aldoximes as imine surrogates. Hydroxamic acids, in general, are known metal chelators successfully applied in analytical chemistry [5], nuclear fuel processing [6], mineral collection [7], corrosion inhibition [8], and drug design. Metal binding properties often determine the biological activity of hydroxamic acids, which can act as siderophores [9] (bacterial iron transporters) as well as inhibitors of metalloenzymes (such as matrix metalloproteases [10], hyston deacetylases [11], and HIV integrase [12]). In one of our previous works, we demonstrated that conjugates of *N*-hydroxy THIQ acids (**1**, R^1^ = OH) with various fluorophores can serve as selective fluorescent chemosensors for specific metal ions [13]. 

In this study, we aimed at developing a modified type of such sensors in which attachment of an external fluorescent probe would be unnecessary if the extended π-system of the probe itself would enable it to act as the source of fluorescence. We envisioned that this could be achieved by the annulation of an additional aromatic ring as in, for example, *5*-hydroxydibenzo[*c*,*h*][1,6]naphthyridine (**2**) (Figure 2). However, while there are a handful of methods for the preparation of 5-alkyl [14] or 5*H*- [15,16] dibenzo[*c*,h][1,6]naphthyridines reported in the literature, the synthesis of the corresponding 5-hydroxy derivatives has not been described. While developing a strategy to construct target compound **2** (the goal which was ultimately achieved), we also arrived, unexpectedly, at a representative of an entirely different heterocyclic system, namely, indolo[3,2-*c*]isoquinolines **3**, which are related to cryptolepine [17], an antimalarial alkaloid from West African plant *Cryptolepis sanguinolenta* [18,19,20]. Our newly discovered approach to the construction of indolo[3,2-*c*]isoquinolines is quite distinct from the methods reported in the literature, most of which are based on the transformations of functionalized indoles [21,22] or condensations of two non-annulated starting materials [23,24,25]. Herein, we present the detailed results obtained in the course of this investigation.

## 2. Results and Discussion

Retrosynthetic analysis of compound **2** with the disconnection of the amide bond brought us to the key intermediate **4** which is based on a typical THIQ scaffold obtainable by the CCR (Figure 3). Since the free amino group would not be compatible with CCR (as it would react with HPA), it was decided to obtain the corresponding nitro derivative **5** first and then reduce it. *O*-Methyl protecting group on hydroxamate moiety was introduced in order to prevent possible elimination of a water molecule in the course of subsequent elaboration of a lactam ring.

Nitro compound **5** was prepared from HPA and *O*-methylated oxime **6** (Scheme 1). The reaction was conducted on a gram scale under solvent-free conditions at 110 °C. Such forcing reaction conditions have been previously shown to be crucial for CCR with poorly reactive *O*-alkyl oximes derived from electron-deficient benzaldehydes [26]. Notably, the yield of CCR product **5** (67%) was substantially higher compared to the yield (36%) obtained in the previously reported reaction of the corresponding unprotected (N-OH) oxime [4]. This increase in yield is likely attributable to the inability of **6** to undergo HPA-mediated dehydration to the respective nitrile, which was the major side reaction for its N-OH counterpart [4]. The *trans-*configuration of compound **5** was based on comparing the *^3^J_HH_* values for vicinal methine protons to those reported in the literature [26].

Surprisingly, the classical reduction of **5** with sodium sulfide (Scheme 1, path a) gave a product with no aliphatic protons in the ^1^H NMR spectrum and only one carbonyl group signal present in the ^13^C NMR spectrum, which did not correspond to the expected structure **4**. The structure of the isolated reaction product, compound **7**, which was found to be a known [27] derivative of indolo[3,2-*c*]isoquinoline, was determined by X-ray diffraction analysis.

The mechanism of transformation **5**→**7** can be rationalized as follows. Presumably, the formation of compound **7** begins with reduction of nitro group giving nitroso intermediate **A**. Under basic conditions (Na_2_S gives strongly alkaline solutions when dissolved in water), **A** can undergo decarboxylation with concomitant trapping of the putative carbanion by the nearby nitroso group, resulting in *N*-hydroxyindoline **B**. Successive elimination of a water molecule (**B**→**C**) and tautomerization (**C**→**D**) could lead to tetracycle **D** which can, under the reducing conditions, produce compound **7** possessing the contiguous aromatic system (Scheme 2). Noteworthy, such a complex cascade of transformations proceeds with a very high yield of compound **7** (91%).

The proposed mechanism can be supported by the following literature data. Firstly, nitroso compounds have been established as intermediates in the reduction of nitroarenes with inorganic sulfides [28]. Decarboxylation of THIQ acids such as **5** is a known process occurring under similarly forcing conditions [29]. Decarboxylation of postulated intermediate **A** could be driven forward by the presence of the nearby nitroso functionality which, in turn, intercepts the carbanion generated upon the loss of CO_2_. Reactions of CH acids with nitroso compounds in the basic medium leading, upon dehydration, to imines have been reported [30,31]. It has also been shown [32] that *O*-alkyl hydroxamic acids can be reduced to amides by molecular sulfur under basic conditions. Molecular sulfur is always present in reaction mixtures containing Na_2_S and an oxidant, in this case, the starting material (**5**).

Several other reducing agents, including sodium dithionite, hydrogen gas+Pd/C, formic acid+Pd/C, ammonium formate+Pd/C, and stannous chloride were tested to perform selective reduction of nitro group in compound **5**. All reagents except for ammonium formate, which gave the desired 2-aminophenyl THIQ acid **4** (Scheme 1, path b) in almost quantitative yield (96%), were unsuccessful (Appendix A). Noteworthy, both types of reduction were performed on a gram scale, and corresponding products **4** and **7** were obtained in analytically pure form with no need for chromatographic purification.

The desired tetracyclic hydroxamic acid **2** based on the dibenzo[*c*,*h*][1,6]naphthyridine core was obtained from THIQ acid 4 in three synthetic steps: lactamization on treatment with propylphosphonic anhydride (T3P) to give compound 8, dehydrogenation of the latter with DDQ and O-demethylation of compound 9 with BBr3. Surprisingly, lactamization 4→8 failed or proved low-yielding with the use of several common amide coupling reagents, including HATU, CDI, DCC, (COCl)2. To the best of our knowledge, compound 2 is the first N-hydroxydibenzo[*c*,*h*][1,6]naphthyridine obtained to-date. Compound 7 was elaborated into 1-chloroisoquinoline 10 on treatment with phosphorus oxychloride. Reduction with zinc in acetic acid gave 11H-indolo[3,2-*c*]isoquinoline **11** which is a known precursor to antiprotozoal compound **12** [33,34] (Scheme 3). Starting from HPA, compound **11** was prepared in an overall yield of 40%, over 4 steps (see Appendix A). This is in sharp contrast to the previously reported synthetic routes to **11** (8% over 4 steps [35] or 12% over 5 steps [36]). Another advantage of the new route to compound **11** presented herein contrasting it to the previously reported approaches [33,34] is its being free from the use of palladium catalysts, which is an obvious upside from the standpoint of pharmaceutical production [37].

## 3. Materials and Methods

### 3.1. General Information

NMR spectra were acquired with a 400 MHz Bruker Avance III spectrometer (400.13 MHz for ^1^H and 100.61 MHz for ^13^C) in DMSO-*d*_6_ and were referenced to residual solvent proton signals (δH = 2.50) and solvent carbon signals (δC = 39.5). Mass spectra were acquired with a Bruker maXis HRMS-ESI-qTOF spectrometer (electrospray ionization mode, positive ions detection). X-ray single crystal analysis was performed on Agilent Technologies «SuperNova» diffractometer with monochromated Cu Kα radiation. The temperature was kept at 293 K during data collection. Using Olex2 [38], the structure was solved with the SHELXT [39] structure solution program using Intrinsic Phasing and refined with the SHELXL [40] refinement package using Least Squares minimization. TLC was performed on aluminum-backed pre-coated plates (0.25 mm) with silica gel 60 F254 with a suitable solvent system and was visualized using UV fluorescence. Flash column chromatography on silica (Merck, 230–400 mesh) was performed with Biotage Isolera Prime instrument. Preparative HPLC was carried out on Shimadzu LC-20AP instrument, equipped with a spectrophotometric detector, column: Agilent Zorbax prepHT XDB-C18, 5 µm, 21.2 × 150 mm. Oxime **6** was prepared according to the procedure in the literature [41]. Homophthalic anhydride, reagents, and solvents were obtained from commercial sources and were used without further purification. All reactions were performed under air.

### 3.2. Synthesis

#### 3.2.1. (±)-(3*S*,4*S*)-2-Methoxy-3-(2-nitrophenyl)-1-oxo-1,2,3,4-tetrahydroisoquinoline-4-carboxylic acid (**5**)

A mixture of homophthalic anhydride (2.2 g, 13 mmol) and 2-nitrobenzaldehyde *O*-methyl oxime **6** (2.4 g, 13 mmol) was thoroughly ground in a mortar and transferred to a screw-cap vial. The reaction mixture was heated at 110 °C for 16 h (oil bath). After cooling to room temperature, the resulting glassy solid was treated with MeOH (20 mL) and sonicated to obtain suspension, which was filtered. The resulting solid was washed with MeOH (10 mL), filtered and dried in air to obtain pure title compound. Yield 3.0 g, 67%, beige solid. ^1^H NMR (400 MHz, DMSO-*d*_6_) δ 13.26 (br.s, 1H), 8.21 (dd, *J* = 8.1, 1.5 Hz, 1H), 8.01 (dd, *J* = 7.5, 1.7 Hz, 1H), 7.64 (td, *J* = 7.6, 1.5 Hz, 1H), 7.60–7.44 (m, 3H), 7.38 (dd, *J* = 7.3, 1.6 Hz, 1H), 7.00 (dd, *J* = 7.8, 1.5 Hz, 1H), 6.33 (d, *J* = 1.3 Hz, 1H), 4.42 (d, *J* = 1.4 Hz, 1H), 3.80 (s, 3H). ^13^C NMR (101 MHz, DMSO-*d*_6_) δ 171.7, 160.8, 147.9, 134.7, 133.7, 133.1, 132.9, 131.0, 130.1, 128.8, 127.9, 127.9, 127.3, 126.6, 61.8, 59.3, 50.2. HRMS (ESI-TOF) *m*/*z* [M + Na]^+^ calcd for C_17_H_14_NaN_2_O_6_ 365.0744, found 365.0748.

#### 3.2.2. 6,11-Dihydro-5*H*-indolo[3,2-*c*]isoquinolin-5-one (**7**) 

To a stirred suspension of compound **5** (1.5 g, 4.4 mmol) in dioxane and water (25 + 25 mL) sodium sulfide nonahydrate (6.3 g, 26 mmol) was added in one portion at room temperature. After heating at 70 °C in an oil bath for 16h the resulting solution was concentrated in vacuo. The residue was suspended in water (50 mL) and filtered. The solid was washed with water (3 × 20mL) and dried in air to provide pure title compound. Yield 1.03 g, 91%, yellow solid. ^1^H NMR (400 MHz, DMSO-*d*_6_) δ 11.85 (br.s, 1H), 8.45 (s, 1H), 8.35 (dd, *J* = 8.1, 1.3 Hz, 1H), 8.17 (d, *J* = 7.9 Hz, 1H), 8.04 (d, *J* = 7.9 Hz, 1H), 7.95–7.75 (m, 1H), 7.61–7.47 (m, 2H), 7.29 (ddd, *J* = 8.2, 7.0, 1.2 Hz, 1H), 7.18–6.99 (m, 1H). ^13^C NMR (101 MHz, DMSO-*d*_6_) δ 161.3, 137.2, 132.9, 129.7, 129.0, 126.1, 125.0, 124.6, 121.4, 119.4, 119.4, 118.9, 117.2, 112.3. HRMS (ESI-TOF) *m*/*z* [M + Na]^+^ calcd for C_15_H_10_NaN_2_O 257.0685, found 257.0684.

Crystal Data for compound **7** C_30_H_20_N_4_O_2_ (*M* = 468.50 g/mol): monoclinic, space group P2_1_/c (no. 14), *a* = 11.0070(3) Å, *b* = 15.9054(4) Å, *c* = 12.3765(4) Å, *β* = 95.846(3)°, *V* = 2155.49(11) Å^3^, *Z* = 4, *T* = 293(2) K, μ(CuKα) = 0.744 mm^−1^, *Dcalc* = 1.444 g/cm^3^, 23901 reflections measured (8.074° ≤ 2Θ ≤ 141.34°), 4127 unique (*R*_int_ = 0.0518, R_sigma_ = 0.0297) which were used in all calculations. The final *R*_1_ was 0.0686 (I > 2σ(I)) and *wR*_2_ was 0.1995 (all data).

#### 3.2.3. (±)-(3*S*,4*S*)-2-Methoxy-3-(2-aminophenyl)-1-oxo-1,2,3,4-tetrahydroisoquinoline-4-carboxylic acid (**4**)

A round-bottom flask was charged with compound **5** (1.4 g, 4 mmol), 10% wt. Pd/C (424 mg, 0.4 mmol), ammonium formate (2.6 g, 41 mmol) and MeOH (150 mL). The resulting mixture was stirred with reflux for 16 h. After cooling to room temperature, it was filtered through a pad of Celite and filtrate was concentrated in vacuo to provide pure title compound. Yield 1.2 g, 96%, beige solid. ^1^H NMR (400 MHz, DMSO-*d*_6_) δ 8.06–7.83 (m, 1H), 7.42–7.33 (m, 1H), 7.34–7.24 (m, 1H), 7.10 (d, *J* = 7.5 Hz, 1H), 7.00–6.78 (m, 1H), 6.69 (d, *J* = 7.9 Hz, 1H), 6.51 (d, *J* = 7.6 Hz, 1H), 6.37 (t, *J* = 7.4 Hz, 1H), 5.78 (s, 1H), 5.13 (br.s, 2H), 3.72 (s, 3H), 3.18 (s, 1H). ^13^C NMR (101 MHz, DMSO-*d*_6_, 353 K) δ 174.1, 161.3, 145.2, 137.4, 131.6, 130.8, 128.3, 127.9, 126.6, 125.1, 124.0, 116.5, 116.1, 74.6, 61.5, 61.2, 52.9. HRMS (ESI-TOF) *m*/*z* [M + Na]^+^ calcd for C_17_H_16_NaN_2_O_4_ 335.1002, found 335.0972.

#### 3.2.4. 5-Methoxy-4b,12-dihydrodibenzo[*c*,*h*][1,6]naphthyridine-6,11(5*H*,10b*H*)-dione (**8**)

To a stirred suspension of compound **4** (100 mg, 0.32 mmol) in dry chloroform (10 mL) a 50% wt. solution of propanephosphonic acid anhydride, T_3_P (410 mg, 0.64 mmol) in toluene was added at room temperature. After stirring for 16 h the reaction mixture was diluted with chloroform and st. NaHCO_3_. The organic layer was separated, washed with water, dried and concentrated to give crude product (85 mg, 90%), which was further purified using column chromatography on silica (eluent CHCl_3_-MeOH, 0–100% of MeOH). Yield 60 mg, 63%, white solid. ^1^H NMR (400 MHz, DMSO-*d*_6_) δ 10.29 (s, 1H), 8.14 (dd, *J* = 7.9, 1.5 Hz, 1H), 8.10 (dd, *J* = 8.0, 1.3 Hz, 1H), 7.83 (d, *J* = 7.9 Hz, 1H), 7.53 (td, *J* = 7.6, 1.5 Hz, 1H), 7.43–7.33 (m, 1H), 7.20 (t, *J* = 7.5 Hz, 1H), 7.04 (td, *J* = 7.7, 1.3 Hz, 1H), 7.00 (dd, *J* = 7.9, 1.3 Hz, 1H), 5.16 (d, *J* = 14.7 Hz, 1H), 4.03 (d, *J* = 14.7 Hz, 1H), 3.90 (s, 3H). ^13^C NMR (101 MHz, DMSO-*d*_6_) δ 167.8, 138.0, 134.9, 132.9, 130.2, 129.4, 128.7, 128.4, 127.8, 125.8, 122.8, 115.7, 79.6, 63.0, 59.7, 45.8 (signal of one carbonyl group is missing and several signals are broadened due to conformational exchange). HRMS (ESI-TOF) *m*/*z* [M + Na]^+^ calcd for C_17_H_14_NaN_2_O_3_ 317.0897, found 317.0912.

#### 3.2.5. 5-Methoxydibenzo[*c*,*h*][1,6]naphthyridine-6,11(5*H*,12*H*)-dione (**9**)

A screw-cap vial was charged with compound **8** (150 mg, 0.51 mmol), 2,3-dichloro-5,6-dicyano-1,4-benzoquinone, DDQ (580 mg, 2.55 mmol) and dry 1,4-dioxane (3 mL). The resulting mixture was heated at 110 °C (oil bath) for 48 h (reaction progress was monitored by TLC). After cooling to room temperature, the reaction mixture was partitioned between ethyl acetate (150 mL) and water, containing ascorbic acid (50 mL + 1 g). The organic layer was separated, washed with water, dried and concentrated to provide pure title compound. Yield 114 mg, 77%, beige solid. ^1^H NMR (400 MHz, DMSO-*d*_6_) δ 12.11 (s, 1H), 9.91 (d, *J* = 8.5 Hz, 1H), 8.90 (d, *J* = 8.6 Hz, 1H), 8.53–8.17 (m, 1H), 7.89 (ddd, *J* = 8.7, 7.1, 1.6 Hz, 1H), 7.74–7.56 (m, 2H), 7.49 (dd, *J* = 8.3, 1.4 Hz, 1H), 7.31 (ddd, *J* = 8.5, 7.0, 1.4 Hz, 1H). ^13^C NMR (101 MHz, DMSO-*d*_6_) δ 160.5, 158.6, 141.9, 139.0, 133.9, 133.5, 131.9, 128.3, 127.7, 127.7, 127.3, 125.7, 122.7, 116.5, 111.3, 105.5, 63.7. HRMS (ESI-TOF) *m*/*z* [M + Na]^+^ calcd for C_17_H_12_NaN_2_O_3_ 315.0740, found 315.0740.

#### 3.2.6. 5-Hydroxydibenzo[*c*,*h*][1,6]naphthyridine-6,11(5*H*,12*H*)-dione (**2**)

A suspension of compound **9** (20 mg, 0.07 mmol) in dry DCM (4 mL) was cooled to 0 °C in an ice bath followed by addition of solution of BBr_3_ (102 mg, 0.41 mmol) in dry DCM (1 mL). The mixture was stirred at 0 °C for 3 h and was then allowed to warm to room temperature. The reaction was quenched with sat. NaHCO_3_ (5 mL) under cooling with ice. The precipitate formed was filtered, washed with water and dried to give crude product (20 mg), which was purified via preparative reverse-phase HPLC eluting with water–acetonitrile (both containing 0.1% TFA). Method: 10 min of 20% ACN, then linear gradient 20–95% ACN for 35 min, flow rate—12 mL/min, column temperature—40 °C. Yield 12 mg, 60%, white glassy solid. ^1^H NMR (400 MHz, DMSO-*d*_6_) δ 12.08 (s, 1H), 12.03 (s, 1H), 9.94 (dd, *J* = 8.6, 1.1 Hz, 1H), 9.18 (dd, *J* = 8.7, 1.3 Hz, 1H), 8.41 (dd, *J* = 7.9, 1.5 Hz, 1H), 7.89 (ddd, *J* = 8.6, 7.1, 1.6 Hz, 2H), 7.70 (ddd, *J* = 8.1, 7.1, 1.1 Hz, 1H), 7.63 (ddd, *J* = 8.3, 7.0, 1.2 Hz, 1H), 7.46 (dd, *J* = 8.3, 1.2 Hz, 1H), 7.26 (ddd, *J* = 8.5, 7.0, 1.3 Hz, 1H). ^13^C NMR (101 MHz, DMSO-*d*_6_) δ 160.7, 159.4, 141.9, 138.9, 133.5, 133.3, 131.9, 129.2, 128.2, 127.6, 127.0, 124.3, 121.8, 116.2, 111.9, 104.1. HRMS (ESI-TOF) *m*/*z* [M + Na]^+^ calculated for C_16_H_10_NaN_2_O_3_ 301.0584, found 301.0584.

#### 3.2.7. 5-Chloro-11*H*-indolo[3,2-*c*]isoquinoline (**10**)

Compound **7** (180 mg, 0.77 mmol) was heated with stirring in POCl_3_ (3 mL) at 100 °C (oil bath) for 16 h in a screw-cap vial. Conversion of starting material was controlled by TLC. After cooling to room temperature, the reaction mixture was carefully poured into ice (25 mL) with vigorous stirring. The resulting solid was filtered, washed with water until neutral pH of washings and dried in air to provide pure title compound. Yield 165 mg, 85%, yellow solid. ^1^H NMR (400 MHz, DMSO-*d*_6_) δ 12.63 (s, 1H), 8.62 (d, *J* = 8.2 Hz, 1H), 8.42 (d, *J* = 8.5 Hz, 1H), 8.19 (d, *J* = 7.9 Hz, 1H), 8.02 (t, *J* = 7.5 Hz, 1H), 7.82 (t, *J* = 7.7 Hz, 1H), 7.72 (d, *J* = 8.2 Hz, 1H), 7.52 (t, *J* = 7.6 Hz, 1H), 7.33 (t, *J* = 7.5 Hz, 1H). ^13^C NMR (101 MHz, DMSO-*d*_6_) δ 142.0, 139.1, 132.1, 131.6, 128.1, 127.9, 127.4, 126.5, 126.0, 124.0, 122.6, 122.1, 120.8, 119.7, 112.6. HRMS (ESI-TOF) *m*/*z* [M + H]^+^ calcd for C_15_H_10_ClN_2_ 253.0527, found 253.0528.

#### 3.2.8. 11*H*-Indolo[3,2-*c*]isoquinoline (**11**) 

A solution of compound **10** (200 mg, 0.79 mmol) in AcOH-water (20 + 2 mL) was heated to 75 °C followed by addition of Zn dust (464 mg, 7.14 mmol) in three portions during 1.5 h (conversion of starting material was monitored by TLC). After cooling to room temperature acid was neutralized with 10% aq. NaOH followed by extraction with DCM (3 × 50 mL). Combined organic layer was dried over Na_2_SO_4_, filtered and concentrated to give crude product, which was purified using column chromatography on silica eluting with DCM-MeOH (20:1). Yield 134 mg, 77%, light yellow solid. ^1^H NMR (400 MHz, DMSO-*d*_6_) δ 11.71 (s, 1H), 9.02 (s, 1H), 8.42 (dd, *J* = 8.3, 1.1 Hz, 1H), 8.26 (d, *J* = 7.8 Hz, 1H), 8.09 (d, *J* = 8.2 Hz, 1H), 7.83–7.68 (m, 1H), 7.65–7.50 (m, 3H), 7.44–7.34 (m, 1H), 7.29–7.18 (m, 1H). ^13^C NMR (101 MHz, DMSO-*d*_6_) δ 145.1, 139.0, 134.0, 130.4, 129.1, 127.8, 127.0, 126.6, 126.0, 123.9, 123.2, 121.6, 120.3, 119.7, 112.3. HRMS (ESI-TOF) *m*/*z* [M + H]^+^ calcd for C_15_H_11_N_2_ 219.0917, found 219.0908.

## 4. Conclusions

In summary, we have expanded the scope of heterocyclic scaffolds accessible though the Castagnoli–Cushman reaction by developing protocols for the switchable reduction of 2-methoxy-3-(2-nitrophenyl)-1-oxo-1,2,3,4-tetrahydroisoquinoline-4-carboxylic acid, leading either to novel tetracyclic hydroxamic acid with dibenzo[*c*,*h*][1,6]naphthyridine core (following lactamization and deprotection) or to indolo[3,2-*c*]isoquinolines (as the result of hitherto unknown cascade heteroannulation reaction). Metal binding properties of dibenzo[*c*,*h*][1,6]naphthyridine-based hydroxamic acid obtained and its ability to mimic bacterial siderophores are currently under investigation in our laboratories. The route to antiprotozoal indolo[3,2-*c*]isoquinolines discovered in the course of this study is the most high-yielding among Pd-free protocols reported in the literature.

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
