# Peer review of "Two Annulated Azaheterocyclic Cores Readily Available from a Single Tetrahydroisoquinolonic Castagnoli–Cushman Precursor"

_molecules, 2020, doi:10.3390/molecules25092049_

Round 1

Reviewer 1 Report

The work reported here by Krasavin and collaborators describes the elegant synthesis of two heterocycles, an indolo[3,2-c]isoquinoline from an unexpected cyclisation reaction and the desired dibenzo[c,h][1,6]naphthyridine, from a common intermediate prepared by Castagnoli-Cushman reaction.

The paper is well-written and documented.  This work gives new and efficient accesses to two interesting heterocycles.

I have the following comments:

1) it would be interesting for researchers and students to add a scheme depicting the mechanism of the Castagnoli-Cushman reaction, as it is not obvious

2) For the reduction of the intermediate, several conditions and reagents have been tried but unsuccessfully. Could the authors precise if they did not observe any reaction with full isolation of the starting molecule, or if decomposition or complex mixture were obtained? A table may be added in the SI with the  different conditions tried and the results.

 3) Concerning the cyclisation step to form the 6 membered ring by lactamization. Considering the stereochemistry of the cyclic intermediate I am not really surprised that lactamization would be difficult due to the distance between the two reacting functions. Did the authors perform any molecular modeling study of this intermediate?

4) Scheme 3, compound 10 is obtained in two steps that should be separated in the scheme as they were not performed in one-pot.  

5) the reference section should be corrected as most journal names are not correctly abbreviated and for ref 8, DOI number is missing

6) SI: the nmr spectra of compounds 9 and 2 clearly show the presence of minor species. It would be necessary to give the corresponding hplc chromatograms.

Author Response

1)      A new figure depicting mechanism of the Castagnoli-Cushman reaction was added -Figure 1 (p.2, lines 35-36). Numbering of other figures was adjusted accordingly.

2)      These data are summarized in Table S1, which was added to ESI, p.S2 and corresponding references were added to manuscript on p.4, line 112 and p.8, line 279.

3)      No, such calculations were not performed. However, the lactamization was achieved with respectable yield.

4)      A new reaction arrow (10→11) was added to Scheme 3 (p.3, line 130) and synthetic steps were separated accordingly

5)      Full journal names were changed to abbreviated in all references except for no.5, since there is no such data available. For ref. 8 there is no DOI available, therefore it was not provided, see publisher’s website for details: https://www.jmaterenvironsci.com/Journal/vol6-7.html

6)      These compounds are at least 95% pure. We are not able to provide the HPLC data at the moment due to the COVID-19 epidemics and our facilities being locked down.   

Reviewer 2 Report

The manuscript describes the unexpected indolo[3,2-c]isoquinoline and dibenzo[c,h][1,6]naphthyridine cyclic systems,via reduction of 2-methoxy-3-(2-nitrophenyl)-1-oxo-1,2,3,4-tetrahydroisoquinoline-4-carboxylic acid. 

The manuscript is clearly and well written and the experiments rightfully described. The compounds were obtained in good yields employed gram scales, which is quite notorious.

The proposed mechanism is quite unusual but well supported by literature.

I only have one concern, which is the follows:

NMR characterization of compound 8: Maybe two carbonyl carbon should appear in 13C NMR. Only one at 167.8 ppm is described. The 13C NMR of compound 8 also shows only one carbonyl carbon. Did the authors have any explanation for this? One aliphatic carbon, besides the OCH3, in the 13C NMR of compound 8, should be overlapped with solvent peaks, the authors should state that in NMR description.

Author Response

Several 13C signals of this compound are broadened, presumably due to conformational exchange occurring in the intermediate time scale. In particular, the aliphatic carbon discussed by reviewer gives a broadened singlet at 45.8 ppm, which does not overlap with solvent signal (corresponding zoomed inset was added to the spectrum in ESI, p.S7). The second carbonyl group signal is indeed missing (also due to broadening) and corresponding comment was added to experimental section (p 6., line 209,210).

Reviewer 3 Report

Heterocyclic compounds are important in many fields, therefore development of novel synthetic methods to access different heterocyclic scaffolds has high applied value. The authors report, besides other, an unexpected experimental outcome resulted in high-yield preparation of indolo[3,2-c]isoqinoline derivative, thus representing a new synthetic method. Perhaps, it would be great to see if the same method could be applied for an extended substrate scope and tested for functional group tolerance beyond the sole example reported. However, the presented results look sufficient for a communication.

The results are clearly presented, experimental part and the supporting information are nicely compiled. All synthesized compounds are properly characterized. 

Specific comments:

  1. Regarding the synthesis of indolo[3,2-c]isoqinolines, it seems that a couple of additional relevant references have been missed from the manuscript: Org. Lett. 2005, 7, 1753 (doi: 10.1021/ol050331m); Tetrahedron Lett. 2011, 52, 1574 (doi: 10.1016/j.tetlet.2011.01.089). Perhaps, introduction (lines 47-51) might also require slight adjustment after citing these works.
  2. Minor remark: page 5, line 152: "performed in air" please correct to "performed under air"

Author Response

1)      Corresponding references were added to the manuscript (p.2, line 50) and reference list (no.24 and 25). All other reference numbers were changed accordingly. Introduction was adjusted accordingly (p.2, lines 49-50)

2) Corrected